# A precision medicine approach to metabolic therapy for breast cancer in mice

Ngozi D. Akingbesote[1,2], Aaron Norman[1,2], Wanling Zhu[1,2], Alexandra A. Halberstam [1,2], Xinyi Zhang [1,2], Julia Foldi [3], Maryam B. Lustberg[3] & Rachel J. Perry [1,2✉]

Increasing evidence highlights approaches targeting metabolism as potential adjuvants to cancer therapy. Sodium-glucose transport protein 2 (SGLT2) inhibitors are the newest class of antihyperglycemic drugs. To our knowledge, SGLT2 inhibitors have not been applied in the neoadjuvant setting as a precision medicine approach for this devastating disease. Here, we treat lean breast tumor-bearing mice with the SGLT2 inhibitor dapagliflozin as monotherapy and in combination with paclitaxel chemotherapy. We show that dapagliflozin enhances the efficacy of paclitaxel, reducing tumor glucose uptake and prolonging survival. Further, the ability of dapagliflozin to enhance the efficacy of chemotherapy correlates with its effect to reduce circulating insulin in some but not all breast tumors. Our data suggest a genetic signature for breast tumors more likely to respond to dapagliflozin in combination with paclitaxel. In the current study, tumors driven by mutations upstream of canonical insulin signaling pathways responded to this combined treatment, whereas tumors driven by mutations downstream of canonical insulin signaling did not. These data demonstrate that dapagliflozin enhances the response to chemotherapy in mice with breast cancer and suggest that patients with driver mutations upstream of canonical insulin signaling may be most likely to benefit from this neoadjuvant approach.

[1] Department of Celullar and Molecular Physiology, Yale University School of Medicine, New Haven, CT, USA. [2] Department of Internal Medicine (Endocrinology), Yale University School of Medicine, New Haven, CT, USA. [3] Department of Internal Medicine (Hematology/Oncology), Yale University School of Medicine, New Haven, CT, USA. ✉email: rachel.perry@yale.edu

**B**reast cancer is the second leading cause of cancer deaths in women, with one in eight women in the U.S. developing this disease over her lifetime. The incidence of breast cancer is projected to increase more than 50% between 2011 and 2030[1]. This troubling prediction is attributable in part to rising rates of obesity, which accelerates the appearance and progression of breast cancer in postmenopausal women[2,3]. Similarly, primary tumor appearance[4–6], volume[5,7–9], and metastasis[7–11] increase with a high-calorie diet in implanted, genetic, and chemically-induced mouse models of breast cancer.

Metabolism-targeting neoadjuvant approaches have gained increasing popularity in breast cancer treatment. The biguanide metformin, an inhibitor of gluconeogenesis[12–14] and the most prescribed diabetes drug worldwide[15], has been by far the most popular antihyperglycemic approach used for cancer. At supra-pharmacologic doses – several orders of magnitude higher than in human patients – metformin is capable of reducing breast cancer cell proliferation[16–19], and slowing tumor growth in vivo in rodents[20–22]. Metformin is safe and shows good target engagement in women with breast cancer. Studies regarding its efficacy, however, have been mixed, with most[23–29], but not all, adjuvant and neoadjuvant studies[30–32] failing to observe differences in tumor cell proliferation markers or in outcomes in human patients.

Preclinical data highlight the newest class of diabetes drug, SGLT2 inhibitors, as a potentially attractive alternative to metformin therapy for breast cancer. As metformin acts primarily to inhibit gluconeogenesis, its ability to lower blood glucose and insulin concentrations is likely minimal under postprandial conditions in individuals without diabetes. SGLT2 inhibitors, by contrast, cause glycosuria and, therefore, glucose wasting under both fasting and postprandial conditions. We recently demonstrated that the SGLT2 inhibitor dapagliflozin slows breast tumor growth when administered as monotherapy in obese mice, and that its anticancer effect was correlated with its ability to reverse fasting hyperinsulinemia[33]. A direct effect of SGLT2 inhibition in tumors cannot be ruled out, because data from the Human Protein Atlas (https://www.proteinatlas.org/about/licence) indicate that breast tumors do express SGLT2, albeit at low levels[34]. However, considering data demonstrating that chronically infusing insulin to negate the insulin-lowering effects of dapagliflozin fully abrogated its beneficial effects, it is most likely that the tumor suppressive effects of dapagliflozin are likely attributable to its ability to reduce insulin-stimulated tumor glucose uptake[33]. However, to our knowledge, SGLT2 inhibitors have never been administered in combination with chemotherapy in in vivo breast cancer models. This is a point of substantial clinical importance because breast cancer treatment regimens are effective, but not unanimously so: response rates are particularly low in estrogen receptor-positive breast cancer[35]. Thus, new neoadjuvant approaches are critical to improve the efficacy of standard-of-care breast cancer treatment.

In addition, at a time when precision medicine approaches are gaining increasing popularity in selecting targeted therapies for patients. Such approaches, however, have largely failed to investigate and target metabolism. In this study, we aimed to generate a genetic signature for breast tumors responsive to dapagliflozin as an adjunct to chemotherapy. In vivo studies in seven murine models of breast cancer with different driver mutations suggested a genetic signature of those tumors that responded to dapagliflozin: in tumors with driver mutations upstream of the insulin receptor, dapagliflozin improved the efficacy of paclitaxel, whereas in tumors with driver mutations downstream of the insulin receptor, dapagliflozin was ineffective. These data predict that tumor genetics may be utilized to design metabolism-targeting neoadjuvant treatments for patients with hyperinsulinemia.

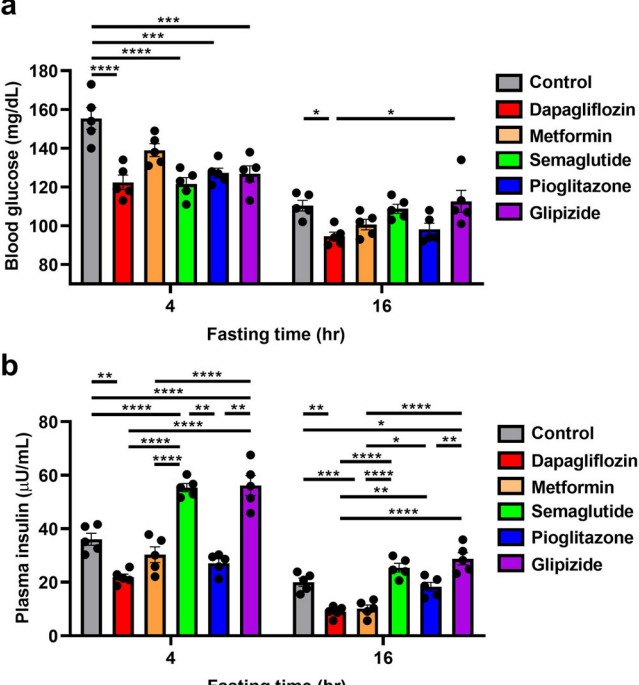

**Fig. 1 The SGLT2 inhibitor dapagliflozin outperforms diabetes drugs in other classes at lowering both fasting and postprandial plasma insulin concentrations.** Mice were fed a Western diet (60% calories from fat in chow, and 5% sucrose drinking water), for four weeks prior to initiation of drug treatment for two weeks. **a** Blood glucose. **b** Plasma insulin. In both panels, *P < 0.05, **P < 0.01, ***P < 0.001, ****P < 0.0001 by ANOVA with Tukey's multiple comparisons test. In both panels, n = 5 per group.

## Results

**Dapagliflozin reduces insulin more than other common diabetes drugs.** Our previous studies have shown plasma insulin concentrations to be a predictor of tumor growth rates in colon[36] and breast cancer[33]. Therefore, we first aimed to determine which commonly prescribed antihyperglycemic medications would most significantly reduce both fasting and postprandial plasma insulin concentrations. The biguanide metformin, the glucagon-like peptide-1 agonist semaglutide, the thiazolidinedione pioglitazone, and the sulfonylurea glipizide all showed some ability to lower plasma glucose concentrations (Fig. 1a). The SGLT2 inhibitor dapagliflozin, however, was the only agent tested to reduce plasma insulin concentrations after both 4 and 16 h of fasting (Fig. 1b). When we measured plasma insulin concentrations in a more detailed time course after 0, 4, 8, 12, 16, and 24 h of fasting, we found that dapagliflozin more robustly lowered plasma insulin area under the curve throughout the day than did metformin (Supplementary Fig. 1a, b). For this reason, we selected dapagliflozin for future studies in murine models of breast cancer.

**Dapagliflozin slows tumor growth in lean and obese MMTV-PyMT mice.** In previous work, we have shown that dapagliflozin slowed E0771 breast cancer growth when applied as monotherapy in obese mice[33]. Here, we examined whether dapagliflozin would slow tumor growth in lean MMTV-PyMT mice, a murine breast cancer model driven by the polyoma virus middle T antigen and commonly expressing an Erbb2 mutation. Although the dose of dapagliflozin utilized for subsequent studies (2.5 mg/kg) was higher than that prescribed in humans (in whom the maximal dose is 10 mg per day), we verified that 2.5 mg/kg dapagliflozin is not so high as to inhibit SGLT1, as indicated by the fact that urine

glucose concentrations in mice treated with 1.0 and 2.5 mg/kg dapagliflozin were not different, as well as the fact that both doses of dapagliflozin caused less significant glycosuria than a non-specific SGLT inhibitor, phlorizin (Supplementary Fig. 2a). Using the standard dose of dapagliflozin employed in this study (2.5 mg/mg), we then demonstrated that it significantly lowered blood glucose (Supplementary Fig. 2b) and insulin concentrations (Fig. 2a), resulting in a reduction in tumor glucose uptake (Fig. 2b) when administered acutely (without differences in body weight). We then moved to continuous dapagliflozin administration. As expected, SGLT2 inhibition caused glucose wasting in urine, suppressing weight gain without causing ketosis (Supplementary Fig. 2c–e). This glucose wasting resulted in chronic reductions in plasma glucose and insulin concentrations in recently fed (6 h fasted) mice given ad lib access to both regular chow and a high-fat, high-carbohydrate Western diet (Fig. 2c, d). These reductions in plasma glucose and/or insulin concentrations inhibited tumor glucose uptake: chronic dapagliflozin treatment lowered MMTV-PyMT tumor [$^{14}$C] 2-deoxyglucose uptake, more markedly than did acute treatment (Fig. 2e). Reduced tumor glucose uptake in dapagliflozin-treated mice correlated with a profound effect of metabolic manipulation on tumor growth: whereas both glucose uptake and tumor growth were accelerated in Western diet fed mice as compared to chow fed, dapagliflozin slowed tumor growth in both lean and obese animals (Fig. 2f). The attenuation of tumor growth was likely due to an effect to slow tumor cell division and not to alter apoptosis, as tumor expression of cytochrome c, an apoptosis marker, was not different in chow and Western diet fed mice treated with dapagliflozin (Supplementary Fig. 2f).

**Dapagliflozin slows 4T1 tumor growth in lean and obese mice.** Next, we studied a second murine tumor model, 4T1 subcutaneous breast cancer, which is driven by a p53 mutation (34,35). BALB/c mice treated with Western diet did become overweight (Supplementary Fig. 3a), and both hyperglycemic and hyperinsulinemic but not ketotic (Fig. 3a, b, Supplementary Fig. 3b). Western diet enhanced both glucose uptake and tumor growth, but dapagliflozin reduced each of these parameters in both chow and Western diet fed mice (Fig. 3c, d). Employing chronic insulin infusion to prevent the effect of dapagliflozin to reduce plasma insulin concentrations, however, abrogated its effect to slow tumor growth in both lean and obese mice. This suggests that dapagliflozin may slow 4T1 tumor growth primarily by its effect to lower circulating insulin concentrations. Next, we examined whether reductions in body weight/adiposity in dapagliflozin-treated mice were likely to explain dapagliflozin's effect to slow tumor growth. To the contrary, supplying physiologic concentrations of saturated fatty acid (palmitate) slowed, rather than accelerated, cell division. Inhibiting fatty acid metabolism by coculturing with etomoxir (validated in Supplementary Fig. 3c) partially rescued the effect of palmitate to slow cell division (Supplementary Fig. 3d). Taken together, these data argue against a direct effect of dapagliflozin to inhibit, or adipose-derived fatty acids to promote, breast cancer growth.

**Dapagliflozin enhances efficacy of chemotherapy in mice with breast cancer.** Although our and others' preclinical data demonstrating an effect of SGLT2 inhibitors to slow tumor growth are promising, dapagliflozin should and will never be administered as monotherapy for breast cancer. We next examined the impact of dapagliflozin in mice treated with standard-of-care paclitaxel. Both paclitaxel and dapagliflozin reduced tumor glucose uptake (Fig. 4a, b) and improved survival (Fig. 4c, d) in

both MMTV-PyMT and 4T1 breast cancer models. However, dapagliflozin and paclitaxel showed synergistic effects against breast cancer: the two drugs showed additive effects to reduce tumor glucose uptake and improve survival in MMTV-PyMT and 4T1 tumor-bearing mice.

**Dapagliflozin does not cause fatigue, weight loss, or neuropathy.** In considering dapagliflozin as a potential neoadjuvant for future study in humans, it is crucial to examine the possibility that it may enhance existing side effects of paclitaxel chemotherapy. To that end, we assessed the impact of dapagliflozin on anorexia and on neuropathy, both common adverse effects of chemotherapy treatment. Encouragingly, we found that dapagliflozin did not reduce food intake or body weight (Fig. 5a–d), cause fatigue as indicated by spontaneous activity (Fig. 5e, f), or alter thermal latency as monotherapy or when added to chemotherapy in either MMTV-PyMT or 4T1 tumor-bearing mice. This demonstrates that dapagliflozin might hold promise as a safe and effective adjuvant to standard-of-care chemotherapy in breast cancer.

**Driver mutations may predict the response to dapagliflozin in breast cancer.** Precision medicine is an increasingly attractive arena. To determine whether certain tumor drivers may be more likely than others to respond to dapagliflozin, we examined the impact of this agent on survival in obese mice with tumors driven by five additional mutation profiles (Supplementary Table 2). The data suggest a genetic signature of tumors responding to SGLT2 inhibition. While dapagliflozin did not synergize with paclitaxel in mice with p53-driven M6, MYC-driven M158 tumors, or MEK1-driven EpH4 1424 tumors with Brca1-driven tumors, it prolonged survival in mice with Pten-driven EMT6 tumors and HRAS-driven Ac711 tumors (Fig. 6a–e), in addition to ERBB2-driven MMTV-PyMT, and p53 and Pik3ca-driven 4T1 tumors (Fig. 4b, d). These effects are unlikely to be due to a direct effect of dapagliflozin on tumor cell division: incubating 4T1 cells in dapagliflozin increased cell number rather than decreasing it, while incubating all of the other tumor cell lines in dapagliflozin had no effect on cell number (Supplementary Fig. 4). Taken together, these data indicate that tumors with mutations upstream of canonical insulin signaling responded to dapagliflozin as an adjuvant to chemotherapy in the current study, whereas tumors driven by mutations downstream of canonical insulin signaling did not (Fig. 7). These data highlight opportunities for the development of precision medicine approaches to utilize metabolic adjunct therapies for breast cancer informed by tumor genetics.

## Discussion
Despite the undisputed epidemiologic link between excess weight and numerous cancers, including breast cancer, clinical trials examining the effect of metabolic therapy (almost exclusively, metformin) against cancer have been under-pursued and underwhelming. This may be at least in part to metformin's modest effect to reduce plasma insulin concentrations by less than 20%[37,38] (Fig. 1b). Given its higher potential efficacy to lower the insulin area under the curve over the course of the day, in this study, we examined the impact of dapagliflozin as an adjunct in combination with chemotherapy. We treated mice with breast cancer driven by a complement of mutations with dapagliflozin in combination with paclitaxel chemotherapy.

Importantly, without enhancing adverse effects of paclitaxel, dapagliflozin improved the efficacy of paclitaxel to promote neuropathy, but it improved the efficacy of chemotherapy to slow tumor growth in murine breast cancer models with mutations in

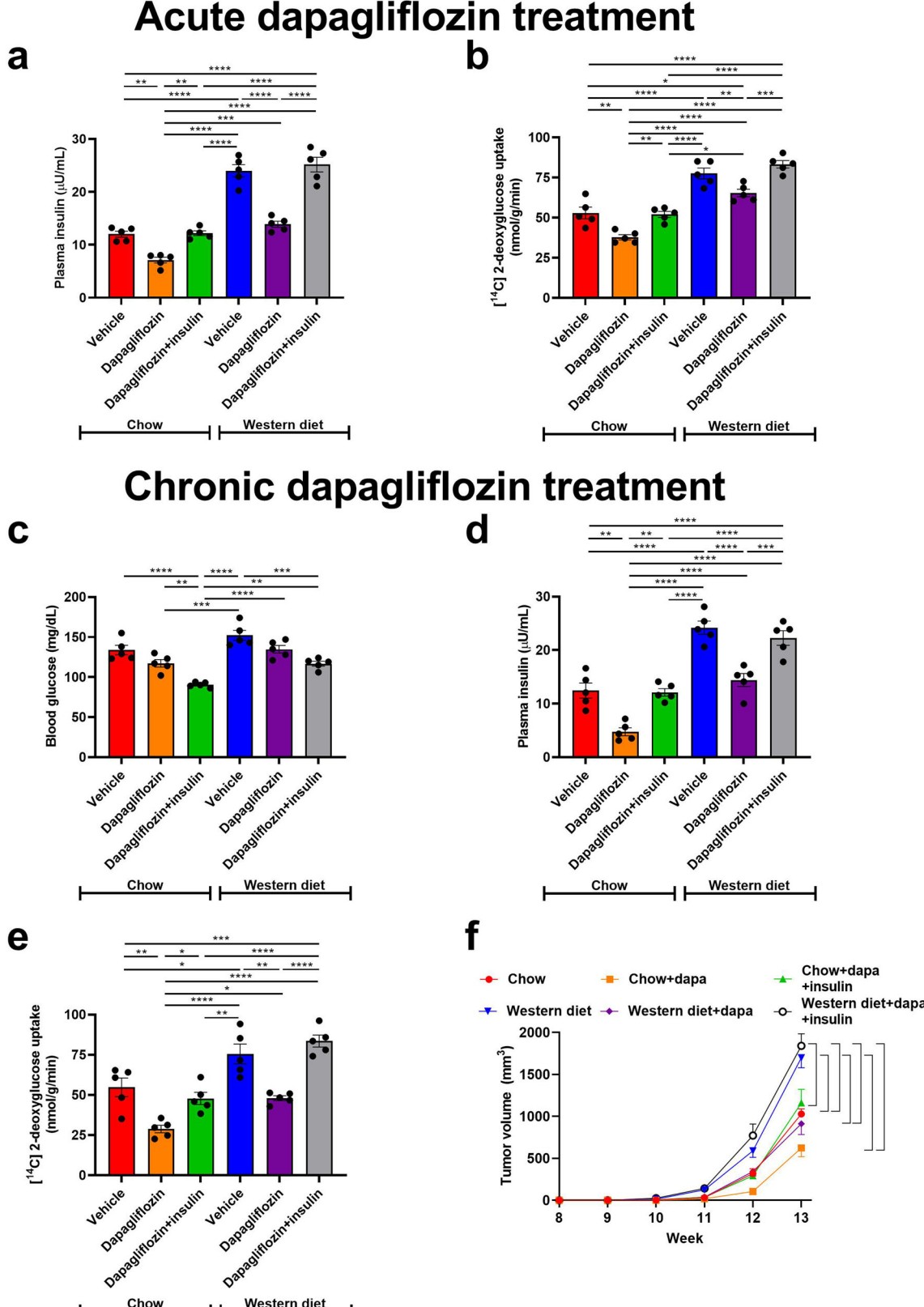

**Fig. 2 Dapagliflozin slows spontaneous tumor growth in lean and obese MMTV-PyMT mice, correlated with its effect to lower plasma insulin concentrations. a** Plasma insulin and (**b**) Tumor [14C] 2-deoxyglucose uptake in mice treated acutely with a single dose of dapagliflozin (2.5 mg/kg). **c**, **d** Blood glucose and plasma insulin in mice treated for four weeks with dapagliflozin in drinking water. **e** Tumor [14C] 2-deoxyglucose uptake in mice treated chronically with dapagliflozin. In panels (**a**–**e**), mice were studied after a 6 h fast. *$P < 0.05$, **$P < 0.01$, ***$P < 0.001$, ****$P < 0.0001$ by ANOVA with Tukey's multiple comparisons test. $n = 5$ per group. **f** Tumor growth. The mean ± S.E.M. of $n = 5$ per group is shown. Brackets denote statistically significant ($P < 0.05$) comparisons by ANOVA with Tukey's multiple comparisons test.

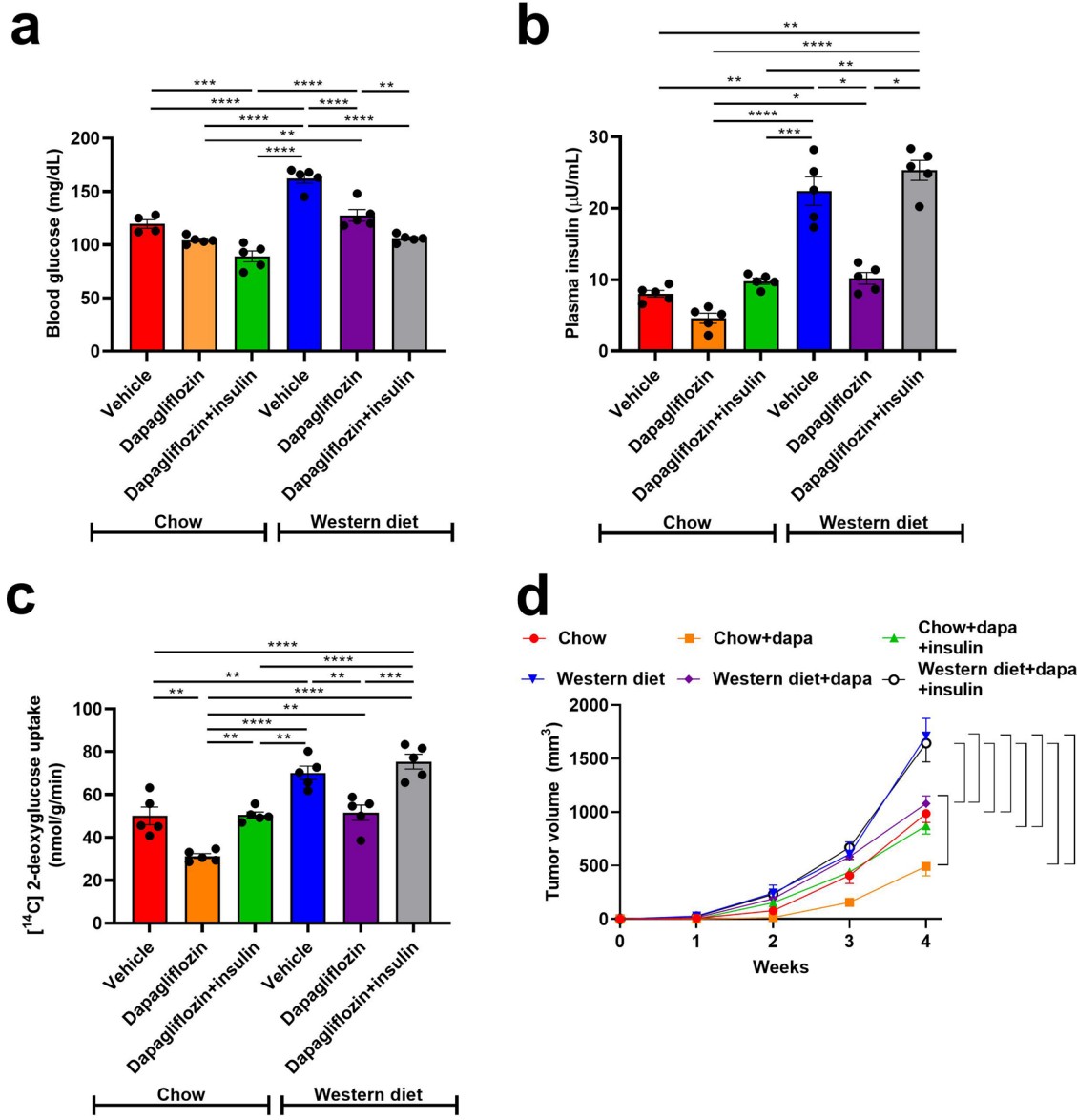

**Fig. 3 Dapagliflozin slows orthotopic 4T1 tumor growth in lean and obese mice, correlated with its effect to lower plasma insulin concentrations.**
**a** Blood glucose and (**b**) Plasma insulin in 6 h fasted mice. **c** Tumor [$^{14}$C] 2-deoxyglucose uptake. In panels (**a–c**), *$P < 0.05$, **$P < 0.01$, ***$P < 0.001$, ****$P < 0.0001$ by ANOVA with Tukey's multiple comparisons test. $n = 5$ per group. **d** Tumor growth. Data are the mean ± S.E.M. of $n = 5$ per group. The brackets denote statistically significant ($P < 0.05$) comparisons by ANOVA with Tukey's multiple comparisons test.

pathways upstream, but not downstream, of canonical insulin signaling (Fig. 7). These data suggest that precision medicine approaches to neoadjuvant treatment are of pressing importance. Indeed, a recent meta-analysis hinted at a differing effect of obesity on outcomes based on cancer subtype: Lohmann and colleagues observed a larger magnitude of the deleterious effect of excess body weight in triple-negative and hormone receptor-positive, HER2-negative breast cancer as compared to hormone receptor-positive, HER2-positive breast cancer[39]. However, these data stop short of suggesting specific breast cancer driver mutations that may be more or less responsive to therapies aiming to reduce circulating insulin concentrations. Therefore, in this study we aimed to generate insights into a genetic signature for responsiveness to dapagliflozin in mice with breast cancer.

The primary advance of the current study is its assessment of which genetic drivers of breast cancer may be most responsive to SGLT2 inhibitors and perhaps other insulin-lowering agents. Our data hint at a genetic signature for tumors in which dapagliflozin

improved the response to chemotherapy: E0771, 4T1, EMT6, and Ac711 tumors are driven by proteins upstream of PI3K/Akt, key mediators of insulin signaling in health[40] and disease[41,42]. Dapagliflozin enhanced the efficacy of paclitaxel in these tumors. By contrast, SGLT2 inhibition did not improve chemotherapy efficacy in tumors driven by mutations in the KRAS/MEK/ERK cell proliferation pathway (M158 and EpH4 1424), or p53 mutations downstream of PI3K/Akt in the cell survival pathway (M6).

Further emphasizing the physiologic relevance of dapagliflozin's inhibition of the insulin signaling pathway, matching plasma insulin concentrations in dapagliflozin-treated 4T1 and MMTV-PyMT tumor-bearing mice to concentrations measured in untreated controls abrogated the beneficial effect of dapagliflozin to enhance the efficacy of chemotherapy in these animals. While we cannot completely rule out a direct effect of dapagliflozin to reduce tumor glucose uptake independently of insulin, two data sets argue against this interpretation. First, we observed no difference in tumor growth rates between untreated controls and mice treated with

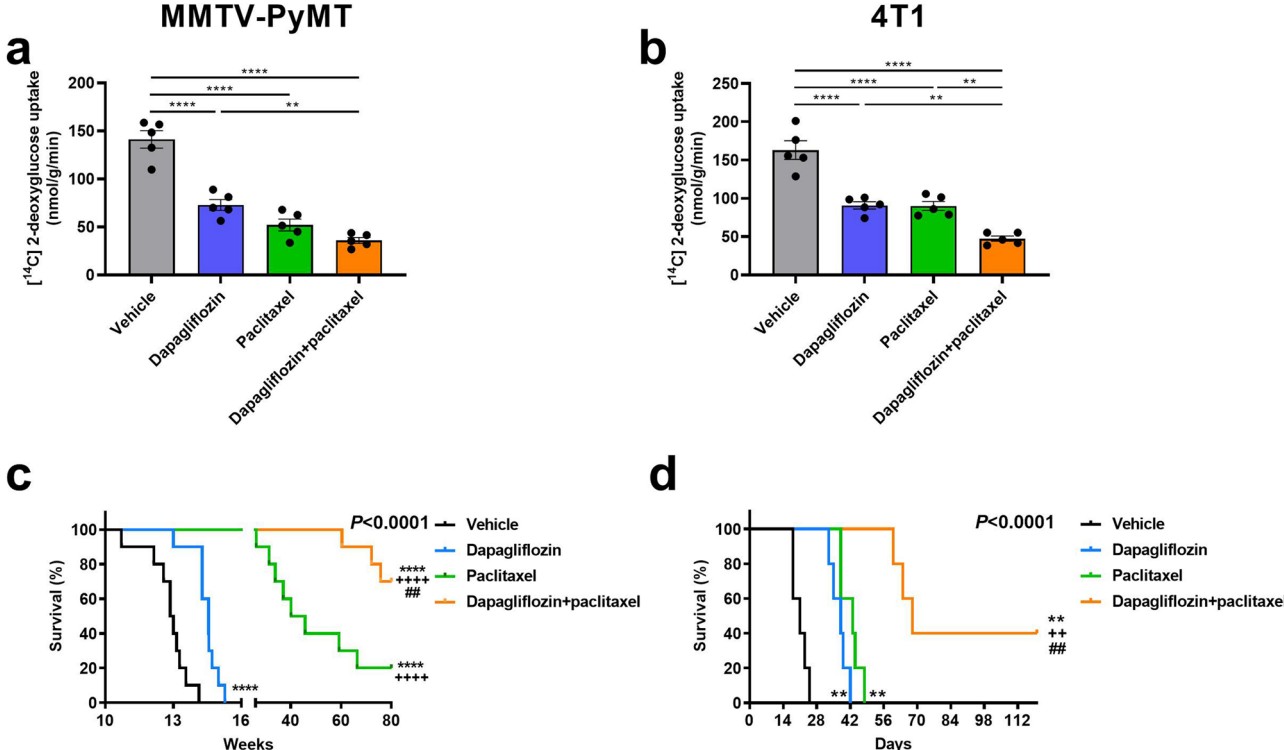

**Fig. 4 Dapagliflozin enhances the efficacy of chemotherapy in two murine models of breast cancer.** Tumor glucose uptake in (**a**) MMTV-PyMT and (**b**) 4T1 tumor-bearing mice. In panels (**a**, **b**), \*\*$P < 0.01$, \*\*\*\*$P < 0.0001$. **c** Survival in MMTV-PyMT mice. The x-axis refers to weeks of life. **d** Survival in 4T1 tumor-bearing mice. The x axis refers to days after tumor cell injection. In panels (**c**, **d**), \*\*$P < 0.01$, \*\*\*\*$P < 0.0001$ vs. vehicle, ++$P < 0.01$, ++++$P < 0.0001$ vs. dapagliflozin, ##$P < 0.01$ vs. paclitaxel via the Mantel-Cox log-rank test, adjusted for multiple comparisons. The p values in the upper right corner of each survival curve refer to the overall curve comparison using the Mantel-Cox log-rank test. Glucose uptake was measured in $n = 5$ per group, and groups of 10 (**c**) and 5 (**d**) were studied to generate the survival curves.

dapagliflozin plus insulin (Figs. 2f, 3d). This suggests that adding insulin undermines the efficacy of the SGLT2 inhibitor in slowing tumor growth. If the SGLT2 inhibitor were acting directly to reduce tumor glucose uptake, insulin infusion would not be expected to fully abrogate the effect of dapagliflozin. Second, the mutational signature of tumors responsive to dapagliflozin – that is, those with mutations upstream of canonical insulin signaling – argues against a direct effect of dapagliflozin on the tumor. If the "direct action" hypothesis were correct, one would predict a similar effect of SGLT2 inhibition to slow tumor growth regardless of whether the driver mutation is related to insulin and/or glucose metabolism.

That said, it is impossible to rule out alternative mechanisms by which dapagliflozin and other metabolism-modulating agents may alter tumor growth. A variety of mechanisms have been proposed to mediate the link between metabolism and cancer, including leptin, adiponectin, resistin, inflammatory cytokines, reactive oxygen species, sex hormones, and others. While probing each of these mechanisms is out of the scope of any single study, it should be noted that dapagliflozin has been shown to modulate expression and/or concentrations of leptin[43], adiponectin[44,45], inflammatory cytokines[43,46], and oxidative damage[47]. Therefore, while the insulin replacement study suggests that dapagliflozin exerts the majority of its effect through reversing hyperinsulinemia, future studies would be required to rule out potential effects of dapagliflozin on each of these other mediators of tumor growth. Additionally, there may be insulin-independent metabolic effects of the combination of dapagliflozin and paclitaxel. While both agents slowed tumor [14C] 2-deoxyglucose uptake as monotherapy, the impact of the combination to reduce tumor glucose uptake was greater than either agent as monotherapy. This suggests that while the underlying anti-cancer mechanisms of the two agents are different (dapagliflozin reduces

systemic glucose and insulin concentrations, which may slow tumor growth by metabolic modulation, and paclitaxel causes microtubule instability), dapagliflozin may sensitize cells to vulnerability to paclitaxel by altering tumor glucose metabolism.

An additional key point in this study is the efficacy of dapagliflozin in lean animals. Although both canagliflozin[33,48,49], dapagliflozin[33], empagliflozin[50], and ipragliflozin[51] have been shown to reduce breast cancer cell division in vitro under high glucose conditions, the efficacy of any SGLT2 inhibitor has rarely been examined in vitro in glucose and insulin concentrations within the physiologic range. Even fewer studies have been performed to examine the efficacy of SGLT2 inhibitors in vivo in lean animals. A single study reported use of dapagliflozin as anti-breast cancer therapy in regular chow fed, athymic, nude mice with MCF-7 breast cancer xenografts[49]. While dapagliflozin was effective in slowing tumor growth in this model, the absence of a typical anticancer immune response in this model leaves questions as to the potential efficacy of this agent in vivo in those with intact immune systems and without obesity.

Relatedly, fasting hyperinsulinemia can occur in those without diabetes and even without impaired glucose tolerance[52] and is rapidly on the rise in the U.S. and worldwide. Between the National Health and Nutrition Examination Surveys conducted between 1988–1994 and 1999–2002, for instance, the age-adjusted incidence of fasting hyperinsulinemia increased by 35%[53]. In the last two decades, the prevalence of hyperinsulinemia even in those without diabetes has increased, while the prevalence of diabetes also increased by close to 50% in the same time period[54]. These data emphasize that there is a critical need to understand therapies that might work for both those with diabetes and those with hyperinsulinemia without diabetes.

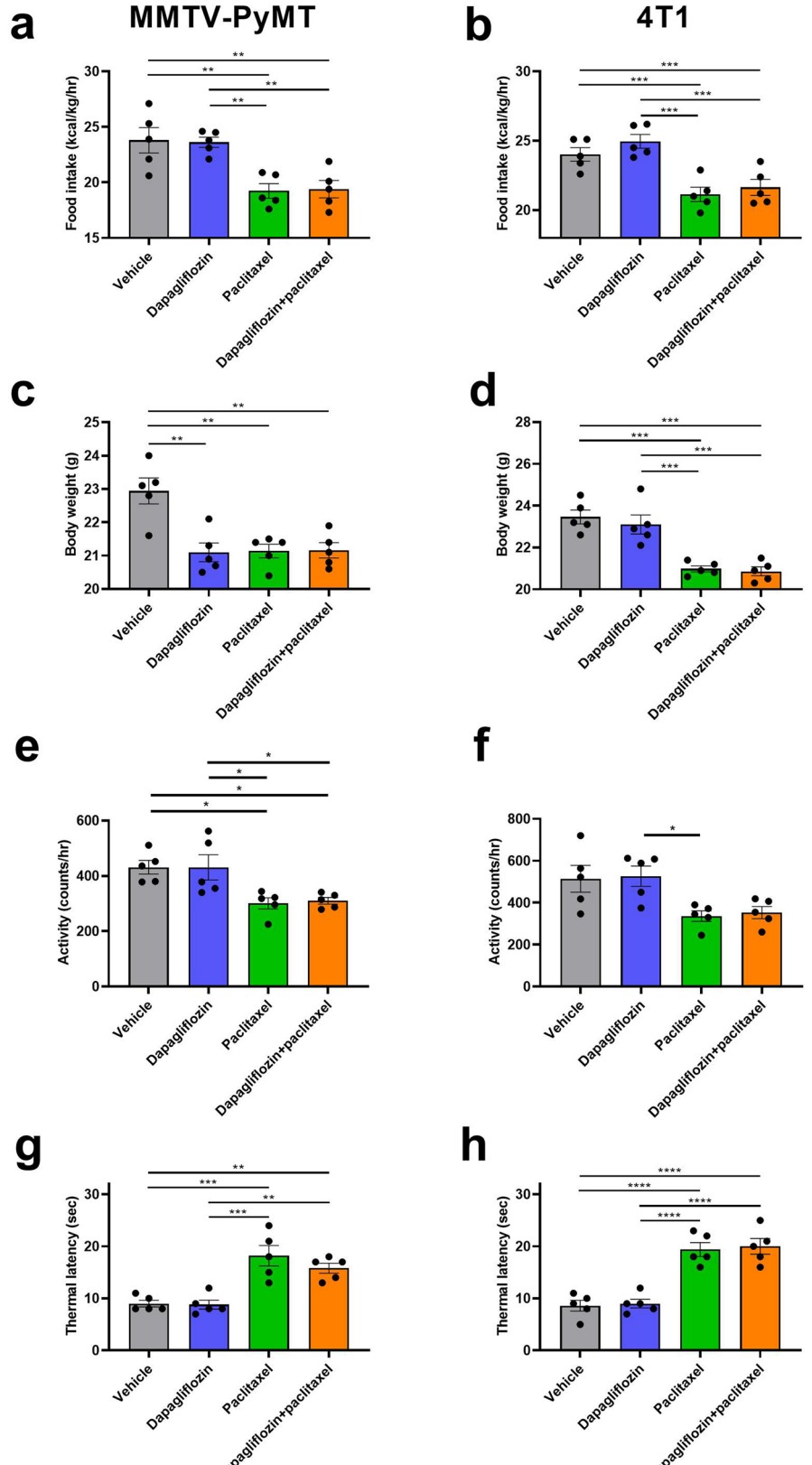

**Fig. 5 Dapagliflozin has no adverse effects to cause fatigue, anorexia, weight loss, or neuropathy in MMTV-PyMT or 4T1 tumor-bearing mice. a**, **b** Ad lib food intake, measured in metabolic cages. All metabolic cage studies in this figure were performed after 3 weeks of chemotherapy or vehicle treatment in MMTV-PyMT and 4T1 tumor-bearing mice, respectively. **c**, **d** Body weight, measured after 4 weeks of treatment. **e**, **f** Spontaneous activity. **g**, **h** Thermal latency during week 3 of chemotherapy or vehicle treatment. In all panels, *$P < 0.05$, **$P < 0.01$, ***$P < 0.001$ by ANOVA with Tukey's multiple comparisons test. Groups of $n = 5$ were studied.

**a** EMT6

**b** Ac711

**c** M6

**d** M158

**e** Eph4 1424

**Fig. 6 Tumor drivers may predict the response to dapagliflozin as an adjunct to paclitaxel. a–e** Survival in EMT6, Ac711, M6, M158, and Eph 1424 tumor-bearing mice, respectively. Mice were examined daily by an investigator who was blinded as to group allocation. **P < 0.01 vs. vehicle, ++P < 0.01 vs. dapagliflozin, ##P < 0.01 vs. paclitaxel by the Mantel-Cox log-rank test, adjusted for multiple comparisons. The P values in the upper right corner of each survival curve refer to the overall curve comparison using the Mantel-Cox log-rank test. Groups of n = 5 were studied.

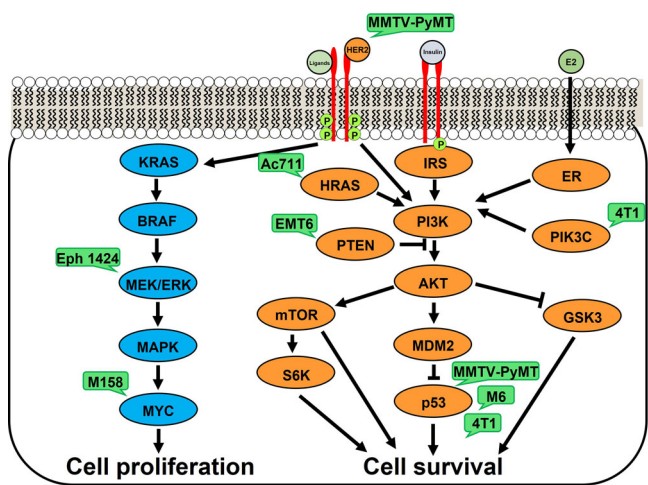

**Fig. 7 Driver mutations tested in this study.** Green bubbles show models utilized herein.

In summary, we demonstrate here that the SGLT2 inhibitor dapagliflozin improves the efficacy of chemotherapy to slow breast tumor growth in a mutation-dependent manner. We reveal that mice with breast cancer driven by mutations upstream of the PI3K/Akt insulin signaling pathway were responsive to dapagliflozin, while those driven by mutations downstream of PI3K/Akt or in pathways with other driver mutations were not. These data support the development of precision medicine, insulin-lowering approaches to breast cancer in hyperinsulinemic patients, with or without diabetes, and position SGLT2 inhibitors as an attractive target to fill this niche.

## Materials and methods

**Study design**. 4T1 (CRL-2539), Ac711 (CRL-3092), EMT6 (CRL-2755), EpH4 1424 (CRL-3071), M158 (CRL-3086), and M6 (CRL-3441) cells were purchased from ATCC and cultured in the manufacturer's recommended media. Cell lines were not authenticated in our laboratory, but are routinely authenticated at ATCC. These cells were injected into mice as described below at passage <10. In the cell division assays, $5.0 \times 10^5$ cells/well were cultured in the manufacturer's recommended media containing 0.5% DMSO and supplemented with palmitate (0.5 mM), etomoxir (0.2 mM), and/or dapagliflozin (100 μM). Cells were counted

by a blinded investigator using a LUNA-II automated cell counter after staining with Trypan blue 48 h later.

To validate the efficacy of etomoxir, we used the method of Antinozzi and colleagues[55]. Briefly, we cultured cells in the manufacturer's recommended media, plus 0.2 mM palmitate in 0.5% DMSO. Etomoxir (0.2 mM) was added to a randomly chosen subset of cells, which were cultured for 48 h, trypsinized, and $1 \times 10^5$ cells were transferred to Erlenmeyer flasks with a holder containing 20 µl of 20% potassium hydroxide and suspending a piece of filter paper above the media. 0.1 µCi [14C] palmitate was added to each flask, the flasks were sealed tightly with a rubber stopper, and shaken for two hours at 37 °C. After two hours, the reaction was stopped by adding 0.5 ml of 1 N perchloric acid, followed by an additional two hours of shaking at 37 °C in order to ensure that all $^{14}CO_2$ was trapped in the filter paper. The filter paper was then removed and counted using the Hidex 300 SL scintillation counter. The counts from each well were normalized to the group mean of counts for the vehicle-treated cells.

All mouse studies were approved by the Yale University Animal Care and Use Committee. To determine the appropriate diabetes drug to lower both fasting and postprandial plasma insulin concentrations, 8–10 week old female C57bl/6J mice were fed a high fat, high carbohydrate Western diet (Research Diets D12492; 60% calories from fat, 20% protein, 20% carbohydrate; 5.21 kcal/g in combination with 5% sucrose drinking water) for four weeks, after which two weeks of antihyperglycemic drug treatment were initiated. Mice were randomized to receive either vehicle (regular water and daily intraperitoneal injections of PBS), metformin (1 mg/ml in drinking water; approximate daily dose 200 mg/kg), dapagliflozin (0.0125 mg/ml in drinking water, approximate daily dose 2.5 mg/kg or, in a small subset of studies, 0.005 mg/ml in drinking water, approximate daily dose 1.0 mg/kg), phlorizin (0.0125 mg/ml in drinking water, approximate daily dose 2.5 mg/kg) semaglutide (50 nmol/kg/day by subcutaneous injection), pioglitazone (10 mg/kg/day by intraperitoneal injection), or glipizide (0.5 mg/ml; approximate daily dose 100 mg/kg). After two weeks, blood glucose and plasma insulin were measured in the same mice after fasting for the duration indicated in the figure legends. Insulin area under the curve (AUC) between each set of data points $t_a$ and $t_b$ (e.g., 0 to 4 h, 4 to 8 h, etc.) was calculated as the area of the trapezoid between these points:

$$AUC = \frac{1}{2} * (Ins_a + Ins_b) * (t_b - t_a).$$ (1)

The sum of the individual areas under the curve from 0 to 24 h represents the total insulin AUC over the course of the day.

For the tumor studies, C57bl/6J, BALB/c, FVB, and MMTV-PyMT mice were purchased from Jackson Laboratories at 7 weeks of age. Upon arrival, a randomly selected subset was given a Western diet, while the remaining mice were given regular chow (ENVIGO-Teklad 2018; 18% calories from fat, 24% protein, 58% carbohydrate; 3.10 kcal/g) and regular water. Water intake was monitored twice weekly for each cage of five mice by weighing water bottles. Two weeks after arrival, mice were injected with tumor cells into the right mammary fat pad (details in Supplementary Table 1). All cell lines were verified pathogen-free by the Yale Comparative Pathology Research Core prior to injection.

All mice – whether they had spontaneous or orthotopic tumors – were monitored twice weekly until they developed palpable tumors, and daily thereafter. Chemotherapy (intraperitoneal paclitaxel, 15 mg/kg twice weekly) was initiated when tumors reached 300 mm³, measured using calipers. A subset of mice in both the chow and Western diet fed groups was randomized to receive dapagliflozin in drinking water beginning the day of chemotherapy induction, with the dapagliflozin concentration adjusted for the measured daily water intake so that each mouse would receive approximately 2.5 mg/kg/day, assuming that all water that disappeared from the water bottles was consumed by the mice. This dose was chosen based on human dosing, scaled for metabolic rate: the maximal human dose of dapagliflozin is 10 mg/day (0.167 mg/day in a 60 kg individual). Whole-body metabolism, as approximated by oxygen consumption per kg body weight, is ~7.5-fold higher in mice as compared to humans (~37.5 L $O_2$ consumption/kg body weight/day in mice, ~5.0 L/kg/day in humans), so that the maximal daily dose when scaled up for body weight is ~1.3 mg/kg/day. The concentration of dapagliflozin was increased to target a dose of 2.5 mg/kg/day to account for water removed from the bottle that was not consumed (i.e., from water dripping or mice licking the water bottle). A subset of otherwise untreated, tumor-bearing mice was subjected to acute dapagliflozin treatment: two doses of dapagliflozin (2.5 mg/kg per dose), each a day apart, by oral gavage. Terminal studies were performed 6 h after the second dose.

In 4T1 and MMTV-PyMT tumor-bearing mice, dapagliflozin-treated mice were further randomized to be implanted with an Alzet pump containing insulin in artificial plasma (115 mM sodium chloride, 5.9 mM potassium chloride, 1.2 mM magnesium chloride hexahydrate, 1.2 mM sodium phosphate monobasic monohydrate, 1.2 mM sodium bisulfate, 2.5 mM calcium chloride dihydrate, 25 mM sodium bicarbonate, 4% bovine serum albumin) or vehicle (artificial plasma). The infusion rates were selected to match plasma insulin concentrations in 4 h fasted dapagliflozin-treated mice to those measured in the diet-matched controls: 18 mU/kg/hr in chow fed mice, and 40 mU/kg/hr in Western diet fed mice. Mice with 4T1 and MMTV-PyMT tumors were monitored, and tumor size measured by a blinded investigator, twice weekly. Body weight was measured weekly. Food intake and activity were measured in

Columbus Lab Animal Monitoring System metabolic cages and averaged over three days during week 3 of treatment. Thermal algesia was assessed by placing mice on a hot plate (IITC Life Sciences) at 52 °C. Thermal latency was defined as the time elapsed before the mouse withdrew its back paw(s), jumped, or scaled the wall of the hot plate enclosure. Mice with Ac711, EMT6, Eph4 1424, and M158 tumors were monitored daily to determine survival. Failure to right was considered a surrogate marker for mortality.

**Flux studies.** One week prior to terminal isotope tracer studies, mice underwent surgery under isoflurane anesthesia to place a catheter in the right jugular vein. Mice were allowed a week of post-surgical recovery, during which they received daily intraperitoneal non-steroidal anti-inflammatory (Carprofen, 5 mg/kg per day) providing analgesic coverage for the first 72 h after surgery. Terminal studies were performed following a 6 h fast. Animals were placed in restrainers and their tails secured lightly using tape, allowing them some space to move around. After 2 h to acclimate to the restrainer, a 1.0 µCi bolus of [2-14C] 2-deoxyglucose was administered through the catheter. Blood was obtained from the tail vein every 10 min following 2-deoxyglucose injection, and blood glucose concentrations were measured using a handheld glucometer. 60 min after radioisotope tracer injection, mice were euthanized using IV Euthasol (50 mg/kg). Tumors were freeze-clamped in tongs prechilled in liquid nitrogen and stored at −80 °C pending further analysis.

Glucose uptake was measured as described previously[56,57]. Briefly, [14C] concentrations were measured with the scintillation counter in plasma (tracer, [14C] 2-deoxyglucose) every 10 min after radioisotope tracer injection, and in tumor ([14C] 2-deoxyglucose-6-phosphate, a trapping pool which cannot be further metabolized) 60 min after injection. The plasma [14C] concentrations were fitted to an exponential decay curve using Microsoft Excel, and the following equation was used for calculation of tumor glucose uptake:

$$Glucose\ uptake = \frac{C_p * C_m^*}{\int_{t=0}^{60} C_p^*(t)\,dt}$$ (2)

where $C_p$ is the average plasma glucose concentration at the 30-, 40-, 50-, and 60-min time points, $C_m^*$ is the tumor [14C] (i.e., [14C] 2-deoxyglucose-6-phosphate) specific activity, and $C_p^*(t)$, used in the integral, is the plasma [14C] 2-deoxyglucose concentration at time t.

**Biochemical analysis.** Blood glucose concentrations were measured using a handheld glucometer, and urine glucose concentrations using the YSI Glucose Analyzer. Plasma insulin concentrations were measured by ELISA (Mercodia), and urine glucose concentrations using the YSI Glucose Analyzer. Plasma β-hydroxybutyrate (β−OHB) concentrations were measured by gas chromatography/mass spectrometry, modifying a protocol we have previously described[58]. Plasma samples were spiked with an equal volume of internal standard [U-13C4] β-OHB (1.0 mM), deproteinized with equal volumes of zinc sulfate followed by barium hydroxide, and centrifuged at 10,000 rpm for 10 min. The supernatant was transferred to a glass gas chromatography/mass spectrometry vial and derivatized with 3 volumes of b-butanol 4 N HCl, after which the samples were heated to 65 °C for 60 min, evaporated under $N_2$ gas, and resuspended in 75 mL of tri-fluoroacetic acid:methylene chloride (1:7). β-OHB concentrations were measured by gas chromatography/mass spectrometry run in chemical ionization mode, with peak areas for $^{12}C$ β-OHB compared to peak areas for $^{13}C_4$ β-OHB at a known concentration.

**Gene expression analysis.** mRNA was extracted in Triazol using the RNeasy Mini Kit (Qiagen). Tumor cytochrome c mRNA expression was measured by qPCR using SYBR Green (Applied Biosystems) using primers (OriGene) with the following sequences:
F: GAGGCAAGCATAAGACTGGACC
R: ACTCCATCAGGGTATCCTCTCC

**Statistics and reproducibility.** Sample sizes were chosen using a power calculation to provide adequate power to detect large differences (as would be required for future clinical trial design) with moderate error. Animals were randomized to experimental groups with the use of a random number generator. All metabolic and tumor size analyses were performed by an investigator who was blinded as to group allocation. Statistical analysis was performed using GraphPad Prism version 9.3.1. Survival analyses were performed using the Mantel-Cox logrank test, adjusting for multiple comparisons. Tumor size, fluxes, and biochemical parameters were compared by ANOVA with Tukey's multiple comparisons test. The mean ± S.E.M. are shown. No data were excluded from analysis. The primary means of verification of reproducibility was the fact that studies were performed in seven separate breast cancer mouse models.

**Reporting summary**. Further information on research design is available in the Nature Research Reporting Summary linked to this article.

## Data availability

Individual data points are shown throughout, except in line graphs where so many data points are present that it would be impossible to pick out individual data points. All raw data are available in Supplementary Data 1.

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

## Acknowledgements

The authors thank members of the Perry lab for helpful discussions, and members of the Yale Animal Resource Center for their care of the mice studied in these studies. This study was funded by awards from the Lion Heart Foundation (to R.J.P.), the National Institutes of Health (T32-GM0007324, supporting NDA), the China Scholarship Council (supporting X.Z.), the Yale Westchester Alumni Association (to A.A.H.), and a Timothy Dwight Richter Fellowship (to A.A.H.).

## Author contributions

The study was designed and the first draft of the manuscript was written by N.D.A. and R.J.P. Experiments were performed and data analyzed by N.D.A., A.N., W.Z., A.A.H., X.Z., and R.J.P. J.F. and M.B.L. provided critical feedback on the study design and manuscript. The final version of the manuscript was edited and approved by all authors.

## Competing interests

R.J.P. has previously received investigator-initiated research funding, for a project related to SGLT2 inhibitors but unrelated to cancer, from AstraZeneca. The remaining authors declare no competing interests.
