## [Peer Review File · Communications Biology]

Reviewers' comments:

Reviewer #1 (Remarks to the Author):

The authors examine the impact of dapagliflozin and paclitaxel in tumor growth. Limited information is available regarding the potential anti-neoplastic effects of the anti-diabetic drug dapagliflozin. This is a topic of great interest given the potential for other anti-diabetic drugs like metformin in cancer treatment. The topic of the study is important, but several points need to be clarified.

- 1) A rationale needs to be provided for the concentrations of the various drugs tested. Are the concentrations relevant in terms of comparisons with plasma concentrations in patients?
- 2) In the abstract, the authors state: "Here we combine the SGLT2 18 inhibitor dapagliflozin with paclitaxel chemotherapy in lean and obese mice". However, there is no data on the impact of paclitaxel+ dapa treatment in obese mice.
- 3) The authors state that dapagliflozin is superior to Metformin as anti-cancer drug due to its insulin-lowering ability. However, metformin was able to lower insulin levels to the same extent as dapa after 16h of fasting. In addition, insulin levels after dapa do not seem to be statistically significant from those after metformin at 4h fasting.
- 4) In figures 2F and 3D, the authors show that administration of insulin to mice treated with dapa accelerated tumor growth, but it is not sufficient to conclude dapa is impairing tumor growth through lowering insulin. It is unclear why reduced adiposity could not explain part of the effect, since adipose tissue produces adipokines that influence insulin signaling (leptin, adiponectin). These mediators are dysregulated in obesity and T2D.
- 5) It is unclear why dapa promotes the growth of 4T1 cells in vitro.
- 6) Control experiments need to be included to demonstrate that etomoxir inhibits FA metabolism.
- 7) The number of tumor models studied does not provide a broad coverage of the insulin signaling pathway and, as such, conclusions pertaining to this need to be softened.

Minor comments:

The insulin signaling pathway should be incorporated in Figure 7.
Referring to each panel of the figure when appropriate instead of citing the whole figure at the end would make the text easier to read.
In lines 204-205, the authors state that breast cancers with mutations downstream (instead of upstream) of insulin signaling are responsive to dapa treatment.
Figure legends could be more descriptive.

Reviewer #2 (Remarks to the Author):

In this study, the authors investigated whether dapagliflozin, a SGLT2 inhibitor, enhances the efficacy of paclitaxel in breast cancer. They observed that combination of dapagliflozin with paclitaxel inhibited tumor growth in lean and obese MMTV-PyMT mice, a murine breast cancer model driven by the polyoma virus middle T antigen and commonly expressing an ErbB2 mutation. Accordingly, they found that dapagliflozin slows cellular growth in 4T1 subcutaneous breast cancer, which is driven by a p53 mutation. Finally, the authors examined the impact of this agent on survival in obese mice with tumors driven by five additional mutation profiles and demonstrated that dapagliflozin prolonged survival in mice with Pten-driven EMT6 tumors and HRAS-driven Ac711 tumors, suggesting that mice with breast cancer driven by mutations upstream of the PI3K/Akt insulin signaling pathway were responsive to dapagliflozin, while those driven by mutations downstream of PI3K/Akt or in pathways with other driver mutations did not. They conclude that precision medicine for breast cancer utilizing dapagliflozin could be useful.

Comments:

- 1) This is an interesting manuscript because the authors demonstrated for the first time that combination therapy of SGLT2i with paclitaxel is beneficial on breast cancer.
- 2) The authors mainly focused on insulin-dependent mechanism of dapagliflozin. Please give a

little more detail about what happened as a result. For instance, how insulin-dependent mechanism affect tumor cells other than growth (e.g. oxidative stress, inflammation)?

3) Is apoptosis involved in reduced tumor growth by dapagliflozin in MMTV-PyMT mice?

4) What is the mechanism of beneficial effect of dapagliflozin plus paclitaxel breast cancer? It is a very interesting observation, but it is unclear what kind of mechanism is expected.

5) Impraglifozlin should be replaced by ipragliflozin

We were delighted to learn that both the editors and reviewers responded positively to our manuscript (COMMSBIO-22-3747A), characterizing it as “of considerable interest,” “a topic of great interest,” and “an interesting manuscript because the authors demonstrated for the first time that combination therapy of SGLT2i with paclitaxel is beneficial on breast cancer.” We appreciate this fair and constructive review and are grateful for the opportunity to respond to the reviewers’ comments (shown in bold) and thereby strengthen this manuscript.

Reviewer #1 (Remarks to the Author):

The authors examine the impact of dapagliflozin and paclitaxel in tumor growth. Limited information is available regarding the potential anti-neoplastic effects of the anti-diabetic drug dapagliflozin. This is a topic of great interest given the potential for other anti-diabetic drugs like metformin in cancer treatment. The topic of the study is important, but several points need to be clarified.

We thank the reviewer for their assessment that this is “a topic of great interest” considering the limited information available regarding the mechanism and effects of dapagliflozin in the setting of cancer.

1) A rationale needs to be provided for the concentrations of the various drugs tested. Are the concentrations relevant in terms of comparisons with plasma concentrations in patients?

We thank the reviewer for raising this important point, which required both textual revisions and a new experiment to address properly. First, we now explain the rationale for the dose administered:

“A subset of mice in both the chow and Western diet fed groups was randomized to receive dapagliflozin in drinking water beginning the day of chemotherapy induction, with the dapagliflozin concentration adjusted for the measured daily water intake so that each mouse would receive approximately 2.5 mg/kg/day, assuming that all water that disappeared from the water bottles was consumed by the mice. This dose was chosen based on human dosing, scaled for metabolic rate: the maximal human dose of dapagliflozin is 10 mg/day (0.167 mg/kg/day in a 60 kg individual). Whole-body metabolism, as approximated by oxygen consumption per kg body weight, is approximately 7.5-fold higher in mice as compared to humans (~37.5 L O₂ consumption/kg body weight/day in mice, ~5.0 L/kg/day in humans), so that the maximal daily dose when scaled up for body weight is approximately 1.3 mg/kg/day. The concentration of dapagliflozin was increased to target a dose of 2.5 mg/kg/day to account for water removed from the bottle that was not consumed (i.e., from water dripping or mice licking the water bottle).”

Additionally, we now provide data to justify that the dose was not excessive. In addition to the fact that dapagliflozin did not induce ketosis (shown in the previous submission, but repeated here as part of our response to this important comment),

we now include urine glucose concentrations in mice treated with one of two doses of dapagliflozin (2.5 mg/kg, the dose used for all other studies in this manuscript, as well as 1.0 mg/kg, more similar to the daily dose used to treat humans) or with phloridzin (a nonspecific SGLT inhibitor). We show that urine glucose concentrations are not different in mice treated with 1.0 and 2.5 mg/kg dapagliflozin, but that both are lower than urine glucose concentrations in mice treated with 2.5 mg/kg phloridzin (left panel below, Supplemental Figure 1A; right panel, SGLT inhibitor treated data from Supplemental Figure 1A, on a linear scale so that differences are not obscured by the scale of the axis):

Taken together, these data demonstrate that the dose of dapagliflozin is not excessive, at least in the sense that it 1) does not inhibit SGLT1, and 2) does not cause ketosis, although we now acknowledge that this dose, per body weight, is higher than that used in humans:

“Although the dose of dapagliflozin utilized for subsequent studies (2.5 mg/kg) was higher than that prescribed in humans (in whom the maximal dose is 10 mg per day)...”

- 2) In the abstract, the authors state: “Here we combine the SGLT2 18 inhibitor dapagliflozin with paclitaxel chemotherapy in lean and obese mice”. However, there is no data on the impact of paclitaxel+ dapa treatment in obese mice.

We apologize for the inaccuracy of this statement as written; as the reviewer states, we treated lean and obese mice, and also combined dapagliflozin with chemotherapy in obese mice. As the reviewer states, we did not treat obese mice with paclitaxel in combination with dapagliflozin, and have edited the abstract accordingly:

“Here, for the first time, we treat lean breast tumor-bearing mice with the SGLT2 inhibitor dapagliflozin as monotherapy and in combination with paclitaxel chemotherapy.”

- 3) The authors state that dapagliflozin is superior to Metformin as anti-cancer drug due to its insulin-lowering ability. However, metformin was able to lower insulin levels to the same extent as dapa after 16h of fasting. In addition, insulin levels after dapa do not seem to be statistically significant from those after metformin at 4h fasting.

We agree with the reviewer’s assessment of the data shown in the prior iteration of our manuscript, and have added new experimental data to clarify. We have now measured the insulin area under the curve throughout 24 hours of fasting, for 24 hours after treatment with dapagliflozin or with metformin, and demonstrate that the insulin area under the curve – which we believe is most important when considering the impact of insulin on tumors, rather than timepoint plasma insulin concentrations – is 30% lower in mice treated with dapagliflozin as compared to metformin:

considered in combination with human trials with metformin that have yielded largely disappointing results, these data highlight the need for alternatives to metformin for targeting breast cancer using metabolism-modulating approaches.

- 4) In figures 2F and 3D, the authors show that administration of insulin to mice treated with dapa accelerated tumor growth, but it is not sufficient to conclude dapa is impairing tumor growth through lowering insulin. It is unclear why reduced adiposity could not explain part of the effect, since adipose tissue produces adipokines that influence insulin signaling (leptin, adiponectin). These mediators are dysregulated in obesity and T2D.

We are glad the reviewer raised this important point. In our view, almost the only way to conclusively prove that dapagliflozin slows tumor growth through lowering hyperinsulinemia would be to knock out the insulin receptor in tumor cells, and show that dapagliflozin does not slow those tumors; however, although we have tried to use siRNA to knock out the insulin receptor, the cells do not divide and thus cannot be implanted. Alternatively, we could manipulate food intake in order to match weight in dapagliflozin-

treated and -untreated mice, but this would present an additional confounder. We therefore believe that the best way to handle this important issue is to soften our conclusions. We have done this in several places in the text (edits in red), for example:

- Abstract: “We show that dapagliflozin enhances the efficacy of paclitaxel, reducing tumor glucose uptake and prolonging survival, **correlated with its effect to reduce circulating insulin**, in some but not all breast tumors.”
- Introduction: “We recently demonstrated that the SGLT2 inhibitor dapagliflozin slows breast tumor growth when administered as monotherapy in obese mice, and that its anticancer effect was **correlated with** its ability to reverse fasting hyperinsulinemia.”
- Results: “This [abrogation of dapagliflozin’s effect to slow 4T1 tumor growth by restoring hyperinsulinemia] suggests that dapagliflozin **may** slow 4T1 tumor growth primarily by its effect to lower circulating insulin concentrations.”

Further, we respectfully draw the reviewer’s attention to several sections in the Discussion in which we carefully avoid stating what we agree we do not prove here, that dapagliflozin works through lowering insulin; however, we assert that our data are consistent with that hypothesis:

- “[D]apagliflozin improved the efficacy of chemotherapy to slow tumor growth in murine breast cancer models with mutations in pathways upstream, but not downstream, of canonical insulin signaling.”
- “The primary advance of the current study is its assessment of which genetic drivers of breast cancer may be most responsive to SGLT2 inhibitors and perhaps other insulin-lowering agents.”
- “Further emphasizing the physiologic relevance of dapagliflozin’s inhibition of the insulin signaling pathway, matching plasma insulin concentrations in dapagliflozin-treated 4T1 and MMTV-PyMT tumor-bearing mice to concentrations measured in untreated controls abrogated the beneficial effect of dapagliflozin to enhance the efficacy of chemotherapy in these animals.”
- “While we cannot completely rule out a direct effect of dapagliflozin to reduce tumor glucose uptake independently of insulin, two data sets argue against this interpretation.”
- “These data support the development of precision medicine, insulin-lowering approaches to breast cancer in hyperinsulinemic patients, with or without diabetes...”

5) It is unclear why dapa promotes the growth of 4T1 cells in vitro.

These data are indeed puzzling. We have now repeated the *in vitro* culture studies with five additional cell lines, and find that the SGLT2 inhibitor promotes only the growth of 4T1 cells, and not any of the other cell lines:

Accordingly, we have adjusted the Results:

“These effects [of dapagliflozin to slow tumor cell division] are unlikely to be due to a direct effect of dapagliflozin on tumor cell division: incubating 4T1 cells in dapagliflozin increased cell number rather than decreasing it, while incubating all of the other tumor cell lines in dapagliflozin had no effect on cell number (Supplementary Figure 4).”

6) Control experiments need to be included to demonstrate that etomoxir inhibits FA metabolism.

We have added the requested experiments. Using ¹⁴C palmitate tracer, we now demonstrate that etomoxir reduces fatty acid oxidation by 55%:

7) The number of tumor models studied does not provide a broad coverage of the insulin signaling pathway and, as such, conclusions pertaining to this need to be softened.

We recognize that seven tumor models do not examine each node of the known breast cancer signaling pathways, and that tumor models do not exist to examine each node of the insulin signaling pathway. Therefore, we have revised to soften these conclusions and simply emphasize the data actually shown in the manuscript, including the following (alterations highlighted in red):

- Abstract: “Our data **suggest** a genetic signature for breast tumors **more** likely to respond to dapagliflozin in combination with paclitaxel.”
- Abstract: “**In the current study**, tumors driven by mutations upstream of canonical insulin signaling pathways **responded** to this combined treatment, whereas tumors driven by mutations downstream of canonical insulin signaling **did not**.”

- Introduction: “*In vivo* studies in seven murine models of breast cancer with different driver mutations **suggested** a genetic signature of those tumors that responded to dapagliflozin...”
 - We did not change the last sentence of the Introduction, but emphasize that it is speculative: “These data predict that tumor genetics may be utilized to design metabolism-targeting neoadjuvant treatments for patients with hyperinsulinemia.”
- Results: “**Driver mutations may predict the response to dapagliflozin in breast cancer.**”
- Results: “**The data suggest** a genetic signature of tumors responding to SGLT2 inhibition.”
- Results: “Taken together, these data indicate that tumors with mutations upstream of canonical insulin signaling responded to dapagliflozin as an adjuvant to chemotherapy **in the current study**, whereas tumors driven by mutations downstream of canonical insulin signaling did not (Figure 7).”
- Discussion: “Therefore, in this study we aimed to **generate insights into** a genetic signature for responsiveness to dapagliflozin in mice with breast cancer.”
- Discussion: “Our data **hint at** a genetic signature for tumors in which dapagliflozin improved the response to chemotherapy: E0771, 4T1, EMT6, and Ac711 tumors are driven by proteins upstream of PI3K/Akt...”
- Discussion: “In summary, we demonstrate here that the SGLT2 inhibitor dapagliflozin improves the efficacy of chemotherapy to slow breast tumor growth in a **mutation-dependent** manner.”

Minor comments:

The insulin signaling pathway should be incorporated in Figure 7.

We thank the reviewer for this excellent suggestion and have added the insulin signaling pathway (circled using a dashed purple line for clarity in the response to reviewers only):

Referring to each panel of the figure when appropriate instead of citing the whole figure at the end would make the text easier to read.

We thank the reviewer for this suggestion and have now referred to each panel of the figure whenever possible (editing the descriptions of Figures 1, 2, 4, 5, Supplementary Figures 2 and 3).

In lines 204-205, the authors state that breast cancers with mutations downstream (instead of upstream) of insulin signaling are responsive to dapa treatment.

We apologize for this error and thank the reviewer for bringing it to our attention. We have corrected this.

Figure legends could be more descriptive.

We have edited to make the figure legends more descriptive, and hope the reviewer will find the edited figure legends to be an improvement.

Reviewer #2 (Remarks to the Author):

In this study, the authors investigated whether dapagliflozin, a SGLT2 inhibitor, enhances the efficacy of paclitaxel in breast cancer. They observed that combination of dapagliflozin with paclitaxel inhibited tumor growth in lean and obese MMTV-PyMT mice, a murine breast cancer model driven by the polyoma virus middle T antigen and commonly expressing an Erbb2 mutation. Accordingly, they found that dapagliflozin slows cellular growth in 4T1 subcutaneous breast cancer, which is driven by a p53 mutation. Finally, the authors examined the impact of this agent on survival in obese mice with tumors driven by five additional mutation profiles and demonstrated that dapagliflozin prolonged survival in mice with Pten-driven EMT6 tumors and HRAS-driven Ac711 tumors, suggesting that mice with breast cancer driven by mutations upstream of the PI3K/Akt insulin signaling pathway were responsive to dapagliflozin, while those driven by mutations downstream of PI3K/Akt or in pathways with other driver mutations did not. They conclude that precision medicine for breast cancer utilizing dapagliflozin could be useful.

Comments:

- 1) This is an interesting manuscript because the authors demonstrated for the first time that combination therapy of SGLT2i with paclitaxel is beneficial on breast cancer.**

We thank the reviewer for this kind comment. We agree!

- 2) The authors mainly focused on insulin-dependent mechanism of dapagliflozin. Please give a little more detail about what happened as a result. For instance, how insulin-dependent mechanism affect tumor cells other than growth (e.g. oxidative stress, inflammation)?**

We agree that other mechanisms may be involved. We believe that our responses to the seventh point raised by reviewer 1, softening our conclusions regarding dapagliflozin's putative action through reducing hyperinsulinemia, partially address this important point raised by reviewer 2 as well. Additionally, we have added a paragraph on this specific point to the Discussion:

“[I]t is impossible to rule out alternative mechanisms by which dapagliflozin and other metabolism-modulating agents may alter tumor growth. A variety of mechanisms have been proposed to mediate the link between metabolism and cancer, including leptin, adiponectin, resistin, inflammatory cytokines, reactive oxygen species, sex hormones, and others. While probing each of these mechanisms is out of the scope of any single study, it should be noted that dapagliflozin has been shown to modulate expression and/or concentrations of leptin (43), adiponectin (44, 45), inflammatory cytokines (43, 46), and oxidative damage (47). Therefore, while the insulin replacement study suggests that dapagliflozin exerts the majority of its effect through reversing hyperinsulinemia, future studies would be required to rule out potential effects of dapagliflozin on each of these other mediators of tumor growth.”

- 3) Is apoptosis involved in reduced tumor growth by dapagliflozin in MMTV-PyMT mice?**

This is an excellent question. We examined the potential role of apoptosis by measuring cytochrome c mRNA expression in tumors, and did not observe any difference with dapagliflozin in either chow or Western diet fed mice:

These data suggest that dapagliflozin likely slows tumor growth through an effect to inhibit cell division, without affecting apoptosis.

4) What is the mechanism of beneficial effect of dapagliflozin plus paclitaxel breast cancer? It is a very interesting observation, but it is unclear what kind of mechanism is expected.

We hypothesize that dapagliflozin’s effect to enhance the efficacy of paclitaxel occurs by inhibiting cell division and thereby sensitizing tumor cells to paclitaxel, which in turn provides a “second hit” to slow cell division. The second hit may be in part metabolic: while both paclitaxel and dapagliflozin reduced tumor glucose uptake as monotherapy, the combination reduced glucose uptake more than either agent alone. We have added a comment to this effect in the Discussion:

“Additionally, there may be insulin-independent metabolic effects of the combination of dapagliflozin and paclitaxel. While both agents slowed tumor [14C] 2-deoxyglucose uptake as monotherapy, the impact of the combination to reduce tumor glucose uptake was greater than either agent as monotherapy. This suggests that while the underlying anti-cancer mechanisms of the two agents are different (dapagliflozin reduces systemic glucose and insulin concentrations, which may slow tumor growth by metabolic modulation, and paclitaxel causes microtubule instability), dapagliflozin may sensitize cells to vulnerability to paclitaxel by altering tumor glucose metabolism.”

5) Impragliflozin should be replaced by ipragliflozin

We apologize for this embarrassing misspelling and have corrected it.

Reviewers' comments:

Reviewer #1 (Remarks to the Author):

The authors have provided an excellent response to my previous comments. The only remaining comment would be to provide information regarding how the fatty acid oxidation results were normalized.

Reviewer #2 (Remarks to the Author):

The authors have adequately revised the manuscript. I have no further comments.

We thank the reviewers for their time spent on this manuscript, and for their constructive reviews, which have substantially enhanced the work. We are pleased that Reviewer 2 finds it ready for publication, and thank Reviewer 1 for stating that “The authors have provided an excellent response to my previous comments.”

The only remaining comment would be to provide information regarding how the fatty acid oxidation results were normalized.

We thank the reviewer for making this point. We have edited the Methods section to provide this information:

Etomoxir (0.2 mM) was added to a randomly chosen subset of cells, which were cultured for 48 hours, trypsinized, and 1×10^5 cells were transferred to Erlenmeyer flasks with a holder containing 20 μ l of 20% potassium hydroxide and suspending a piece of filter paper above the media...The counts from each well were normalized to the group mean of counts for the vehicle-treated cells.